# Developing a decision instrument to guide abdominal-pelvic imaging of blunt trauma patients: Methodology and protocol of the NEXUS abdominal-pelvic imaging study

Ali S. Raja[1][☯], Robert M. Rodriguez[2][☯], Malkeet Gupta[3,4], Eric D. Isaacs[2], Lucy Z. Kornblith[5], Anand Prabhakar[6], Noelle Saillant[7], Paul J. Schmit[8], Sindy H. Wei[9], William R. Mower[10][☯]*

1 Department of Emergency Medicine, Massachusetts General Hospital, Harvard Medical School, Boston, Massachusetts, United States of America, 2 Department of Emergency Medicine, San Francisco General Hospital, UCSF School of Medicine, San Francisco, California, United States of America, 3 Department of Emergency Medicine, Ronald Reagan UCLA Medical Center, UCLA Geffen School of Medicine, Los Angeles, California, United States of America, 4 Antelope Valley Hospital Emergency Department, Lancaster, California, United States of America, 5 Department of Surgery, San Francisco General Hospital, UCSF School of Medicine, San Francisco, California, United States of America, 6 Department of Radiology, Massachusetts General Hospital, Harvard Medical School, Boston, Massachusetts, United States of America, 7 Department of Surgery, Massachusetts General Hospital, Harvard Medical School, Boston, Massachusetts, United States of America, 8 UCLA Department of Surgery, Ronald Reagan UCLA Medical Center, UCLA Geffen School of Medicine, Los Angeles, California, United States of America, 9 UCLA Department of Radiological Sciences, Ronald Reagan UCLA Medical Center, Los Angeles, California, United States of America, 10 UCLA Department of Emergency Medicine, Ronald Reagan UCLA Medical Center, Los Angeles, California, United States of America

☯ These authors contributed equally to this work.
* wmower@ucla.edu

**Data Availability Statement:** No datasets were generated or analyzed during the current study. All

## Abstract

Although computed tomography (CT) of the abdomen and pelvis (A/P) can provide crucial information for managing blunt trauma patients, liberal and indiscriminant imaging is expensive, can delay critical interventions, and unnecessarily exposes patients to ionizing radiation. Currently no definitive recommendations exist detailing which adult blunt trauma patients should receive A/P CT imaging and which patients may safely forego CT. Considerable benefit could be realized by identifying clinical criteria that reliably classify the risk of abdominal and pelvic injuries in blunt trauma patients. Patients identified as "very low risk" by such criteria would be free of significant injury, receive no benefit from imaging and therefore could be safely spared the expense and radiation exposure associated with A/P CT. The goal of this two-phase nationwide multicenter observational study is to derive and validate the use of clinical criteria to stratify the risk of injuries to the abdomen and pelvis among adult blunt trauma patients. We estimate that nation-wide implementation of a rigorously developed decision instrument could safely reduce CT imaging of adult blunt trauma patients by more than 20%, and reduce annual radiographic charges by $180 million, while simultaneously expediting trauma care and decreasing radiation exposure with its attendant risk of radiation-induced malignancy. Prior to enrollment we convened an expert panel of trauma surgeons, radiologists and emergency medicine physicians to develop a consensus

relevant data from this study will be made available upon study completion.

**Funding:** The authors received no specific funding for this work.

**Competing interests:** The authors have declared that no competing interests exist.

**Abbreviations:** A/P, abdominal/pelvic; CI, confidence interval; CT, computed tomography; ED, emergency department; NEXUS, National Emergency X-Radiography Utilization Study; NPV, negative predictive value; SAPI, significant abdominal-pelvic injuries.

definition for clinically significant abdominal and pelvic injury. In the first derivation phase of the study, we will document the presence or absence of preselected candidate criteria, as well as the presence or absence of significant abdominal or pelvic injuries in a cohort of blunt trauma victims. Using recursive partitioning, we will examine combinations of these criteria to identify an optimal "very low risk" subset that identifies injuries with a sensitivity exceeding 98%, excludes injury with a negative predictive value (NPV) greater than 98%, and retains the highest possible specificity and potential to decrease imaging. In Phase 2 of the study we will validate the performance of a decision rule based on these criteria among a new cohort of patients to ensure that the criteria retain high sensitivity, NPV and optimal specificity. Validating the sensitivity of the decision instrument with high statistical precision requires evaluations on 317 blunt trauma patients who have significant abdominal-pelvic injuries, which will in turn require evaluations on approximately 6,340 blunt trauma patients. We will estimate potential reductions in CT imaging by counting the number of abdominal-pelvic CT scans performed on "very low risk" patients. Reductions in charges and radiation exposure will be determined by respectively summing radiographic charges and lifetime decreases in radiation morbidity and mortality for all "very low risk" cases.

**Trial registration:** Clinicaltrials.gov trial registration number: NCT04937868.

## Introduction

Injuries to the abdomen and pelvis can lead to significant morbidity and mortality in patients with blunt trauma [1, 2]. Due to this potential for injury, pelvis x-rays are recommended by the American College of Surgeons Committee on Trauma's Advanced Trauma Life Support course as an integral part of the preliminary evaluation of patients with blunt trauma [3]. In addition to pelvis x-ray and focused assessment with sonography in Trauma (FAST) scanning, abdominopelvic (A/P) CT imaging has become the most common emergency department (ED) imaging modality of the abdomen; its use has more than doubled over the past decade [4, 5].

This increase in imaging may be due to the insensitivity of physical exam findings in patients with blunt trauma, especially patients with altered mental status [6–17]. A recent large meta-analysis found that while there were a number of physical examination findings that increased the likelihood of intra-abdominal injury, no individual sign could reliably rule out injury [16]. Due to the insensitivity of physical examination, some authors have advocated liberal A/P imaging of patients with blunt trauma [8–12, 18–21]. Pelvic x-rays and FAST scanning have shown inadequate sensitivity for detecting injuries, leaving A/P CT imaging as the definitive means of assessing injury status [16]. However, there are no definite recommendations that detail which patients with blunt trauma should receive A/P CT imaging [16, 22].

It is important to note that while there has recently been a significant increase in the use of A/P CT among patients with blunt trauma, there has not been a corresponding decrease in the mortality of these patients, including those with significant abdominal-pelvic injuries (SAPI) [23, 24]. In addition, recent studies have demonstrated that selective CT imaging and serial examinations during a period of observation may negate the need for CT in many patients with blunt trauma [25–27]. The apparent lack of true clinical benefit of the increased imaging is especially striking given the risks associated with A/P CT radiation exposure; for every 725 trauma patients between 20 and 40 years of age imaged with an A/P CT, one will develop lethal

cancer from the CT alone [28]. In addition to this lack of proven clinical benefit and potential patient harm, there are likely to be significant cost savings associated with more judicious application of A/P imaging [29].

It is clear that clinical decision instruments designed to guide A/P imaging are necessary, and should follow the design of similar instruments developed and validated for head, cervical spine, and chest imaging in patients with blunt trauma [30–35]. Previous investigators have proposed guidelines for abdomen-specific decision instruments, but these do not offer guidance on the use of combination A/P CTs, the form of imaging that is routinely used in trauma evaluations, and also rely on laboratory analyses–which typically provide results well after imaging decisions have been made and imaging completed [8, 9]. There is also guidance for pelvis x-rays from a large single-center study using criteria from patient history and physical examination [17]. However, this rule has yet to be prospectively validated in a multicenter study, and is not suitable for making decisions on A/P CT.

While adults make up the majority of patients with trauma, similar work has been performed in the pediatric population. In fact, the publication of a recently developed and validated multicenter decision instrument guiding the use of A/P CT in children with blunt trauma serves to highlight the feasibility and need for a similar decision instrument in adults [36]. In this multicenter study we seek to develop and validate a decision instrument to identify adults with blunt trauma who are at very low risk of significant abdominal-pelvic injuries, and for whom the use of A/P CT may be unnecessary.

## Objectives

### General design

In this prospective, cohort, multi-center study we seek to develop an imaging decision tool that will enable clinicians to identify patients with blunt trauma who are at "very low-risk" for clinically significant A/P injuries. Implementation of the decision tool will require clinicians to assess the presence or absence of specific clinical findings. Patients with a full assessment of all criteria and who exhibit none of the clinical findings will be designated as "very low-risk". Patients who exhibit one or more of the clinical findings, as well as those for whom a complete assessment of criteria cannot be completed, will be designated as "not low-risk." Patients who are designated as very low-risk will have negligible potential to harbor significant A/P injuries, will receive no benefit from A/P CT imaging, and will therefore be safe to exclude from A/P CT imaging. All other patients remain candidates for imaging. It is worth noting that while our project focuses on developing a rule for combined A/P CT imaging, the rule will also be suitable for evaluating patients for isolated abdominal or isolated pelvic imaging. This is a reflection of the fact that patients who are designated as low risk by our instrument will have negligible risk of either abdominal or pelvic injury, and can safely be excluded from abdominal-pelvic imaging, isolated abdominal imaging, or isolated pelvic imaging.

Classification by an optimal decision instrument will reflect a balance between the risk of imaging (malignant transformation from exposure to ionizing radiation), the costs of CT, and the risks of not imaging (potential harm from unrecognized significant injuries). Because the rate of lethal malignant transformation is low, the decision tool must exhibit high sensitivity to ensure that the missed injury rate is very low, and that virtually all patients with significant injuries are designated as "not low risk."

To be clinically useful, the decision tool must also assign low-risk classification to as many uninjured patients as possible. This is equivalent to requiring the tool to produce as few false positive results as possible, or equivalently, exhibit high specificity. The proportion of patients

who can safely be spared imaging is directly related to the instrument's specificity, so the development process will focus on identifying a tool with the highest possible specificity.

A third aspect that must be addressed in the development process is the applicable patient cohort, which is in turn related to the intent of the rule. This is an issue that is often inadequately addressed in the development of decision instruments, but can have a profound effect on the ultimate utility of these rules. In situations where occult disease is prevalent, it may be necessary to develop a rule that applies to a broad cohort, including populations with low disease prevalence and low suspicion of disease. In addition to guiding when not to image, these instruments, sometimes referred to as two-way rules, additionally implicitly inform providers of which patients *need* to be imaged in order to reliably detect patients with occult injuries. This approach typically produces many false positive results, and can easily lead to increased imaging, which may be justified by the rule's ability to reliably identifying occult presentations.

An alternative approach is to develop decision instruments that inform providers of which patients *do not need* imaging. This approach is favored when occult presentations are rare and providers are faced with large numbers of patients who exhibit findings that might indicate presence of injury, while the proportion of patients who actually have significant injuries is small.

In blunt trauma, the existing literature indicates that occult A/P injuries are exceptional, and that most patients with significant injuries either exhibit evidence of their injury, or exhibit characteristics that preclude a reliable assessment (e.g., intoxication or altered mentation) [23, 24]. The current status quo is that A/P CT is always or nearly always ordered for blunt trauma patients who truly have injuries, and very few abdominal/pelvic injuries remain undetected. However, along with this high detection rate of A/P injury, clinicians are also imaging a large number of patients who do not have injuries, resulting in very low yield rates of A/P CT. It is in this population that a highly sensitive decision instrument could prove of value by identifying patients who do not have injuries and who *do not need* imaging.

For this study, as with the prior NEXUS studies, we thus focus on patients who have already been identified as higher risk by the fact that the treating providers are considering ordering CT imaging. Our study cohort thus consists of patients with blunt trauma whose presentation is sufficiently concerning to merit further evaluation with A/P CT imaging, and the ultimate goal of the project is to create a decision rule that can safely decrease CT utilization among this population. We will thereby generate a *one-way rule* to be applied to patients whom clinicians were likely going to image with CT–the instrument will reliably detect all patients who have clinically significant injuries, and conversely, patients who have very low risk of injury in whom they can forego CT [37].

This approach is predicated on the assumption that occult A/P injury presentations are rare, and while the existing literature supports this idea, this assumption needs to be verified to address the potential for work-up bias. Thus, an additional component of the study is to complete an evaluation of patients with blunt trauma who do not receive A/P CT to determine how often (if ever) they are subsequently found to have significant A/P injuries that were missed on their initial assessment.

## Materials and methods

### Defining primary outcomes via expert panel review

Previous experience has shown that clinicians differ in their tolerance for missing minor injuries and in their definitions as to what constitutes a clinically significant injury. Many clinicians are comfortable using tools that occasionally miss injuries that require no intervention,

provided all major injuries are identified. Other clinicians seek to identify all injuries regardless of their clinical significance [38].

To accommodate varied perspectives on clinically significant injury, we employed a modified Delphi process and convened an expert panel consisting of three trauma surgeons, three emergency medicine physicians and radiologist to define the following study outcome classifications of injuries seen on CT: clinically significant major injury, minor injury, and insignificant injury.

Our final outcome classification scheme is based in results of this Delphi process among this expert panel. Under this schema, injuries of major clinical significance consist of all abdominal and pelvic injuries requiring intervention, as well as any injury to the aorta, and any injury to the spine associated with instability or neurological compromise. Injuries that require only observation, but no intervention, will be considered clinically minor, provided they do not involve the aorta or spine, while injuries that required neither intervention nor observation will be considered insignificant. See Table 1.

## Core methods

Our basic approach involves collecting assessments on the presence or absence of specific individual candidate criteria, as well as a definitive outcome assessment (presence or absence of A/P injuries) for individual patients with blunt trauma in a large cohort. We will examine different combinations of these criteria to identify a subset that predicts the presence of injury with high sensitivity, while simultaneously exhibiting the highest possible specificity. Our proposed rule will consist of the criteria identified by this process, and risk designation will be based on the presence or absence of the individual criterion, with "low-risk" categorization assigned to individuals who exhibit none of the criteria, and "not low-risk" categorization assigned to those who exhibit one or more of the criteria.

The process used to construct our decision instrument has the potential to create a rule that exhibits superb performance for patients in our derivation cohort, but much lower performance when applied to a new patient cohort. To address this concern, we will conduct a validation study that will evaluate our proposed rule on a new cohort of patients with blunt trauma to determine whether it retains adequate sensitivity to justify clinical application. The potential for decreased A/P CT imaging will be reflected in the specificity observed in the validation phase. The study, along with a condensed summary and list of participating centers, is registered in ClinicalTrials.gov as NCT04937868.

This study has been reviewed and approved by the UCLA Institutional Review Board (UCLA IRB). The study has been granted a waiver of informed consent as it does not alter the care and poses no harm to enrolled patients, and would be impossible to conduct without the waiver.

## Study sites and subjects

This prospective cohort study will be conducted at four Level 1 Trauma Centers, with wide variations in geographic location and patient populations. Including institutions from different environments (urban suburban, rural) enables us to increase the external validity of the instrument and is also necessary to assemble the large cohort of patients needed to obtain high levels of precision, small confidence intervals, and robust measures of sensitivity and negative predictive value that are needed to justify clinical application [8, 9, 11, 12, 16–21, 39, 40].

We will enroll patients using a prospective convenience sample of consecutive patients presenting to the ED between 0700–2300 (the times during which research assistants will be available). We will review ED logs and identify patients who are not enrolled, and we will compare

**Table 1. Classification of injuries based on Delphi consensus.**

**Major injuries**

Bladder or ureteral injury requiring intervention

Bowel injuries requiring intervention

Diaphragmatic injuries requiring intervention

Gynecological injuries requiring intervention

Hepatobiliary injuries requiring intervention

Hip fractures requiring intervention

Male genital injuries requiring intervention

Pancreatic injuries requiring intervention

Pelvic fractures (major—excludes minor avulsion injuries and non-displaced ring fractures) requiring intervention

Pelvic fractures (minor) requiring intervention

Renal injury requiring intervention

Retroperitoneal injuries requiring intervention

Spinal injuries (unstable or with neurological compromise) needing observation or intervention

Spinal injuries (stable with no neurological compromise) requiring intervention

Splenic injury requiring intervention

Vascular injury (aortic) needing observation or intervention

Vascular injury (pelvic vessels) requiring intervention

Vascular injuries (other vessels) requiring intervention

**Minor injuries**

Bladder or ureteral injuries needing observation, but not requiring intervention

Bowel injuries needing observation, but not requiring intervention

Diaphragmatic injuries not requiring intervention (observation status is irrelevant)

Gynecological injuries needing observation, but not requiring intervention

Hepatobiliary injuries needing observation, but not requiring intervention

Hip fractures needing observation, but not requiring intervention

Male genital injuries needing observation, but not requiring intervention

Pancreatic injuries needing observation, but not requiring intervention

Pelvic fractures (major—excludes minor avulsion injuries and non-displaced ring fractures) not requiring intervention (observation status is irrelevant)

Pelvic fractures (minor) needing observation, but not requiring intervention

Renal injury needing observation, but not requiring intervention

Retroperitoneal injuries needing observation, but not requiring intervention

Spinal injuries (stable with no neurological compromise) needing observation, but not requiring intervention

Splenic injury needing observation, but not requiring intervention

Vascular injury (aortic) that do not need intervention or observation

Vascular injury (pelvic vessels) needing observation, but not requiring intervention

Vascular injuries (other vessels) needing observation, but not requiring intervention

**Insignificant injuries**

Bladder or ureteral injuries that do not need intervention or observation

Bowel injuries that do not need intervention or observation

Gynecological injuries that do not need intervention or observation

Hepatobiliary injuries that do not need intervention or observation

Hip fractures that do not need intervention or observation

Male genital injuries that do not need intervention or observation

Pancreatic injuries that do not need intervention or observation

Pelvic fractures (minor) that do not need intervention or observation

Renal injuries that do not need intervention or observation

(*Continued*)

**Table 1.** (Continued)

| |
|---|
| Retroperitoneal injuries that do not need intervention or observation |
| Spinal injuries (stable with no neurological compromise) that do not need intervention or observation |
| Splenic injuries that do not need intervention or observation |
| Vascular injuries (pelvic vessels) that do not need intervention or observation |
| Vascular injuries (other vessels) that do not need intervention or observation |

the age, sex and injury severity scores of the enrolled and un-enrolled populations. We will adhere to the Standards for the Reporting of Diagnostic accuracy studies (STARD) guidelines for diagnostic accuracy studies and include a STARD flow diagram [41].

## Inclusion/Exclusion criteria

The study will be observational by design and will not alter the care or management of blunt trauma patients. Medical management will be determined by treating physicians using current standards of care.

To reduce the potential for bias, the study will seek to enroll all victims of blunt injury who undergo A/P CT imaging as part of their ED trauma evaluation. This may include adult patients of all ages, including the elderly, any and all races, both sexes, women who are pregnant or have childbearing capacity, and any other representative demographic or social groups that may present among patients with blunt injuries. An individual will become eligible for the study when the treating physician determines that A/P CT imaging is needed as part of their trauma evaluation. This eligibility criterion aligns with our goal to develop instruments that will be used to safely decrease CT imaging–the fact that clinicians have already decided to order imaging in enrolled patients *en face* supports the notion that our instruments may only serve to decrease CT ordering.

There will be no exclusion criteria. However, while the study will seek to enroll all patients with blunt injury who undergo A/P CT imaging, treating physicians will be able to waive data collection and immediately obtain imaging on any patient they feel is unstable or in whom they cannot complete their study evaluation prior to imaging. These patients will be designated as "unstable," with stability being an implicit criterion of the rule. Physicians will be requested to complete data entry for these patients at their earliest opportunity, ideally before imaging results are known. This process will not threaten the study validity, however, because instability will be an *a priori* criterion, while blood pressure and heart rate parameters consistent with instability are, themselves, candidate criteria and may become part of the final decision instrument.

## Subject identification and recruitment

Any patient with a blunt injury undergoing emergency A/P CT imaging will be identified as a study subject. Cases will be formally identified when the treating physician requests A/P CT imaging. At this point, the ordering clinician will be approached by a study research assistant and asked to complete a survey that documents the presence or absence of individual risk criterion. The research assistant will also record the patient demographic and identifying information needed to complete the enrollment.

## Human subjects considerations and procedures

The study will collect data from the routine examination and evaluation of patients with blunt trauma. Every patient presenting with blunt trauma will initially undergo a clinical evaluation.

The examining physicians will determine whether imaging is indicated based upon existing institutional treatment protocols and/or individual physician clinical judgment. There will be no study criteria to perform or abstain from imaging any patient, and such decisions will be left entirely to the discretion of the treating physicians. Because the study collects information from routine examinations, does not alter the care of individual patients, and will not retain or release identifying information on any individual, it meets the criteria for exemption from informed consent. The study has completed human subjects institutional review and been granted exemption of informed consent through the UCLA IRB and at the participating centers.

## Data collection

Our collected data will include patient demographic information (date, time, age, sex, and race) and the presence or absence of each individual study variable. Each record will also carry embedded information indicating the identity of the institution producing the data. Because we want to increase the external validity of the study to reflect general practice, we will not provide rigid definitions for any of the individual data elements. This will help ensure that the validated low-risk criteria are widely applicable; clinicians will be asked to judge whether these elements are present on the basis of their routine clinical assessment, as they would when eventually applying the decision instruments. However, for purposes of clarification, we will provide clinicians with general descriptions of each clinical characteristic. These will be presented as informational material to all sites and will be reviewed with clinicians during in-service training sessions conducted before the beginning of the study. Potential predictor variables have been selected according to *a priori* determined associations with SAPI and acceptable inter-rater reliability [8, 17, 21, 42–45]. Additional demographic, historical, vital sign, imaging, and outcome data will also be collected (S1 File presents our data collection instrument for candidate criteria).

Our study will also collect A/P CT imaging and outcome results for each patient. All patients will have received A/P CT imaging, and may have received plain pelvic imaging. These initial studies may be supplemented by any other imaging studies deemed necessary by clinicians involved in the patient's care.

Clinical radiologist at each site will review the x-ray and CT images and prepare final radiographic reports. We will review the final radiographic interpretations to determine whether an individual has sustained a SAPI, but will classify injuries as major, minor, or insignificant based on review of procedural records.

## Phase 1: Derivation of optimal low-risk decision instrument(s) for A/P CT imaging

The first step in identifying suitable risk criteria is to assess the reproducibility of the assessments on the presence or absence of each individual criterion. This is achieved by having paired physicians performed independent evaluations of each criterion on patients undergoing A/P CT imaging, and comparing the results of the two assessments for each individual criterion.

For our study, each of two paired physicians will independently determine whether an individual patient exhibits any of the following potential predictor characteristics: 1) abdominal pain or tenderness, 2) flank pain or tenderness, 3) pelvic pain or tenderness, 4) hip or iliac pain or tenderness, 5) midline lumbar spine or sacral pain, 6) abdominal distention, 7) abdominal or pelvic bruising, 8) abdominal or pelvic abrasion, 9) evidence of genitourinary trauma, 10) abnormal alertness, 11) evidence of intoxication, 12) distracting painful injury, 13)

hypotension, 14) tachycardia, 15) unstable vital signs, 16) low hemoglobin or hematocrit, 17) falling hemoglobin or hematocrit, 18) dangerous mechanism of injury, 19) abnormal plain radiography. For analytic purposes, each variable will be considered negative (normal) unless it can be assessed and found to be abnormal. Thus, variables that cannot be measured in an individual patient (such as pain in an unconscious individual) will be documented as "unable to assess" and treated statistically as negative. This type of imputation has the effect of ensuring that the final derivation process is based on actual observed criteria, and while this may decrease the observed sensitivity of the derivation rule (some patients may have exhibited criteria that were not assessed and been classified as not low-risk if the criteria had been assessed), it leads to a more robust derivation.

We will employ Scott's pi statistic to assess physician inter-rater reliability of the independent assessments of each criterion and designate those exhibiting a point measure of 0.40, or greater, as exhibiting substantial inter-rater agreement and suitable for inclusion in the decision instrument. S1 File contains a copy of the survey instrument we will use to record criterion assessments.

We will assign a final clinical outcome for each patient based on the results of A/P CT imaging as well as the patient's hospital course. Patients who we designate as having injuries must have injuries evident on CT imaging. This reflects the fact that if injuries are not apparent on CT imaging, it is immaterial whether the patient underwent such imaging. Furthermore, we will classify each patient as having, in order of increasing severity: 1—no injuries (no injuries evident on CT); 2—insignificant injuries (e.g., minor bruising, small hematomas, or isolated small avulsion fractures); 3—clinically minor injuries (significant injuries that require no specific intervention); and 4—clinically major injuries (significant injuries that require specific intervention or result in patient death). Our final outcome classification will be based on the most severe type of injury sustained by each patient. We will concatenate our final outcome assignments with the criterion assessments to create our final analysis database.

We will apply chi-squared binary recursive partitioning to this data set to identify separate combinations of criteria that identify both major and minor significant abdominopelvic injuries with > 98% sensitivity and > 98% NPV, assuming such combinations exist. If more than one combination of variables exhibits the requisite sensitivity and negative predictive values for injury, the combination with the highest specificity will be adopted as the optimal low-risk decision instruments.

## Phase 2: Validation of low-risk decision instruments for A/P CT imaging

At the completion of the derivation phase, we will begin enrolling patients into a validation study to assess, and attempt to validate, the test characteristics of our newly developed decision instruments. We will record the presence or absence of the individual criteria of the new decision tool(s), and we will combine these assessments with results of A/P CT imaging and clinical course to form our validation data set. We will evaluate criteria in the validation phase using methodology similar to that used in the derivation phase, although two aspects of assessment will change between the two phases. The first change reflects the reduced number of number of criteria assessments that need to be completed on each patient in the validation phase, as clinicians will only need to assess criteria included in the derived rule(s). A second important difference between the two phases involves the handling of un-assessed criteria. In the validation phase, and in the ultimate application of the rule, each criterion will be considered positive (abnormal) unless it can be assessed and found to be normal. Thus, variables that cannot be measured in an individual patient (such as pain in an unconscious individual) will be documented as "unable to assess" and treated analytically as positive. This assures that low-

risk status, and exclusion from imaging, is based on actual observations of normal findings, and prevents low-risk assignment on the basis of inadequate or incomplete assessment.

Clinical outcomes and injury assessments in the validation phase will be identical to those completed in the derivation phase.

We will assign risk status to each patient based on the criteria assessments, and use these risk assessments in combination with outcome assignments to calculate point measures and 95% confidence intervals for sensitivity, negative predictive value, specificity, and estimated potential reductions in imaging that might be achieved by implementation of each of the rules (both minor and major injuries).

## Work-up bias assessment

In concordance with prior NEXUS instruments, including the NEXUS C-spine, Head CT, and Chest CT rules [30, 33, 35, 46], our A/P decision tool is intended to be applied only to patients who are felt to have sufficient risk of SAPI to merit imaging. The tool is not intended for application on all patients with blunt trauma, particularly those who are determined, on the basis of clinical judgment, to have minimal risk of abdominopelvic injury and for whom A/P CT imaging is not ordered. Restricting application of the tool among low risk patients decreases the potential for generating false positive assessments, which could increase the rate of unnecessary imaging among low risk patients and markedly decrease the utility of the tool.

The success of this approach hinges on the reliability of clinical judgment, and requires that clinicians exhibit high sensitivity in their imaging decisions. High sensitivity for clinical judgment implies that clinicians will obtain imaging for virtually all injured patients, and that missed and occult injuries are rare. Therefore, an important component of our methodology is to assess the potential for unrecognized injuries among patients with blunt trauma and verify that unrecognized and occult injuries truly are rare.

Validating this approach, and verifying the high sensitivity of clinical judgment, is equivalent to assessing the potential for work-up bias, and requires a specific focus on assessing the prevalence of SAPI among patients with blunt trauma who do not undergo A/P CT imaging during their initial ED evaluation.

To conduct this assessment, we will have research assistants, present in the ED on a continuous basis, identify eligible patients during their initial visit and include them in our work-up bias study. We will complete criteria assessments for each patient at the time of presentation and follow each patient for a period of three months to determine whether at any time during this interval they returned for further care or evaluation or sought care at another facility after their discharge, whether they received any x-rays or imaging studies, and if so, what type, and whether they received any treatments, hospital admission, surgery, radiographic intervention, or died as a result of the injury.

We will classify patients as having had a missed SAPI if they underwent a surgical or interventional radiographical procedure, or died as a result of their injury during the 3- month observation period, provided they did not sustain an additional traumatic event in the interim. We expect that there will be some patients who experience multiple traumatic events (e.g., falls in the elderly) where it may be difficult to identify the exact event associated with their injuries. For the purposes of our work-up bias assessment, we will classify injuries as related to the initial traumatic event unless there is compelling information to conclude otherwise (e.g., a patient with a hip fracture who is able to ambulate between events, but becomes incapacitated after a second fall). This conservative approach is intended to overestimate, rather than underestimate, the potential work-up bias.

To form a complete data record for each patient, we will combine injury assessments with initial survey results. We will then classify each patient into one of the following three

categories: 1—No missed or occult injuries; 2—Missed injury, but the patient exhibited one or more of the predictive criteria from the decision tool; or 3—Occult injury, where the patient sustained an injury but exhibited none of the decision criteria. We will report point measures and 95% confidence intervals for the prevalence of each of these outcomes among our follow-up cohort, and we will specifically use the proportion of patients with occult injuries to estimate the potential for work-up bias.

## Outcome ascertainment

As mentioned previously, we will review final radiographic interpretations to determine the presence or absence of SAPI for each patient, but will classify injuries as major, minor, or insignificant based on review of procedural records. Because we are focused on assessing the need for imaging, injuries relevant for our study must be evident on A/P CT imaging. The risk classification assigned by our instrument is immaterial if an injury is not visible on CT imaging, as the injury will not be detected by such imaging.

Because there are differing opinions on the need to diagnose all injuries versus only major clinically significant injuries, we will develop two related instruments. The first will be an instrument that is highly sensitive in detecting major injuries: a goal that will improve specificity and improve the rule's ability to decrease imaging. The tradeoff of this approach is that the rule may miss some minor injuries, and therefore be unacceptable to some clinicians.

To accommodate this perspective, we will also develop an instrument exhibiting high sensitivity in detecting all injuries, including clinically minor injuries. This instrument will likely exhibit lower specificity and a lesser ability to decrease imaging, but will enable us to provide a decision tool that will serve all clinicians regardless of their perspectives on injury detection.

Incorporating this modification enables us to provide refined definitions for our primary and secondary objectives. The primary goal of the study is to develop a decision instrument that reliably identifies patients who harbor injuries of major clinical significance that are evident on A/P CT imaging. The secondary goal of the study will be to develop a decision instrument that reliably identifies patients who harbor injuries of major or minor significance that are evident on A/P CT imaging.

We will consider all injuries noted to be acute by the interpreting radiologists be concerning A/P injuries. Study investigators will review the medical records of any ambiguous reports and, using information from all available radiologic studies and interventional procedures, determine the final injury classification while blinded to any information about clinical variables—thus without knowledge of the patient's classification on the basis of the decision instruments. Table 2 contains the data form we will use to collect injury information. Injury information will be concatenated with clinical data to form the final study database. While the determination of the presence or absence of SAPI will be based on the final interpretation of the imaging studies, the need for medical intervention will be based on whether the patient died as a consequence of the SAPI, or underwent an intervention related to an injury in the abdomen or pelvis during their acute care.

## Global assessment of potential missed injuries

For patients enrolled in the study and admitted as inpatients, hospital discharge summaries will be reviewed in order to determine whether any SAPI were missed by the initial ED evaluation. We will also review trauma, surgery and mortality logs, and quality improvement records to identify any cases of SAPI that were missed on initial presentation (including patients enrolled in the work-up bias assessment), and whether the involved patients underwent A/P CT imaging.

**Table 2. NEXUS abdominal/pelvic CT imaging injury assessment form.**

| Patient Study Number: | | | | |
| --- | --- | --- | --- | --- |
| Please check one, and only one box for each row to provide your classification of the associated injury | | | | |
| | CLASSIFICATION[a] | | | |
| INJURY DESCRIPTION | No Injury | Clinically Insignificant Injury | Clinically Minor Injury | Clinically Major Injury |
| Bladder or Ureteral Injury | | | | |
| Bowel Injury | | | | |
| Diaphragmatic Injury | | | | |
| Gynecological Injury (ovaries, fallopian tubes, uterus, vaginal canal) | | | | |
| Hepatobiliary Injury | | | | |
| Hip Fracture | | | | |
| Male Genital Injuries | | | | |
| Pancreatic Injuries | | | | |
| Pelvic Fractures–Major[b] | | | | |
| Pelvic Fractures–Minor[c] | | | | |
| Renal Injury | | | | |
| Retroperitoneal Injuries | | | | |
| Spinal Injuries–Unstable or with neurological compromise | | | | |
| Spinal Injuries–Stable and with no neurological compromise | | | | |
| Splenic Injuries | | | | |
| Vascular Injuries–Aorta | | | | |
| Vascular Injuries–Pelvic Vessels | | | | |
| Vascular Injuries–Other Vascular Injuries | | | | |

[a]Classification—Injury categories

No injury = No injury evident on A/P CT imaging

Clinically Insignificant injury = Unimportant injury that could be missed without significant consequences

Clinically Minor = Injury that may not require therapy, but is important to diagnose

Clinically Major = Injury that is associated with diagnostic and therapeutic implications

[b]Excludes minor avulsion injuries and non-displaced anterior ring fractures

[c]Minor fractures and non-displaced anterior pelvic ring fractures

**Interventions include:** Surgical intervention; Fracture reduction, repair or stabilization; Interventional radiology procedures or embolizations; Administration of blood or blood products; Reversal of anticoagulation; Administration of pressors

## Data storage and confidentiality

Temporary study index numbers will be assigned to each patient in order to link radiographic results and procedural interventions to survey data via the following procedure. At the time a radiographic request is made, the patient's medical record number and study index number will both be recorded by a research assistant. The information in this file will be only available to the site principal investigators, and will be password protected. This file will contain no additional patient data or identifying information. The survey information for each patient will be stored in a separate secure file, and will be indexed by the assigned study numbers. This file will not contain any identifying patient information or links to individual patients. Once the radiographic and interventional results have been collected for a given patient, the study coordinators will examine the computer data file containing patient medical record numbers and ascertain the corresponding study index number. Radiography reports and outcomes will then be copied and labeled with the study index number, effectively redacting all identifying information from the report. The relabeled photocopies and outcomes will then be forwarded

to a central repository for concatenation with the study survey data. At the completion of the study, the data linking each patient's medical record and study index number will be deleted from each site's encrypted database.

## Data analysis

The final concatenated database will be reviewed at the close of the study. Cases lacking radiographic reports or intervention outcomes will be deleted from the final data analysis.

**Formulation of the optimal A/P CT decision instruments.** As stated previously, the primary focus of the study is to develop a decision instrument that detects injuries of major importance. Recognizing that a decision instrument that misses minor injuries will be unacceptable to some clinicians, a secondary goal of the study is to develop a decision instrument that detects all significant injuries.

To achieve these goals, we will use binary recursive partitioning to construct a combination of reliable criteria for A/P CT imaging that predicts SAPI with > 98% sensitivity, excludes significant injuries with > 98% NPV, and retains the greatest sensitivity. Each individual criterion will be viewed as a dichotomous variable. These criteria will then be used to construct the nodal points of the recursive partitions. Sensitivity will be used as the outcome measure for constructing the partitions.

We will perform two separate recursive partitioning analyses. The first partitioning will use major injury as a final outcome, while the second partitioning will focus on identifying patients with major or minor injuries. Our partitioning processes will continue until one of the following three end-points is reached. Case 1—We have isolated a subset of our criteria that identify all patients in our cohort who have significant injuries. By construction, the criteria identified in this process will have 100% sensitivity in detecting injury while retaining the highest discriminating capacity under the chi-squared analyses. In this situation we will regard the partitioning process as successful and the identified criteria will form our decision instrument(s). Case 2—We exhaust all criteria in the partitioning, but are unable to isolate a subset that captures all injured patients. In this event, we will examine the sensitivity of the combined criteria, and will regard the partitioning as successful if the lower confidence limit for the sensitivity exceeds 98%. 3—We exhaust all patients in the cohort. In this event, a limited number of criteria will predict all injuries but will not be able to discriminate between injured and uninjured patients (i.e., sensitivity of 100%, but specificity of 0%). In this case we will sequentially remove potential predictor variables (beginning with the last identified criterion) and examine the sensitivity of the remaining variables. We will regard the partitioning as successful if the lower confidence limit for sensitivity of the remaining variables exceeds 98%. We will regard the partitioning as unsuccessful if we cannot meet the lower limit threshold.

**Validation of the A/P CT decision instruments.** The validation phase of the study will involve the collection of the combination of clinical variables forming the A/P CT decision instruments [17]. A case will be considered true-positive if the patient has at least one of the variables and is given a diagnosis of SAPI based on the designated imaging modality. Cases of patients given a diagnosis of SAPI but in whom none of the variables are present will be classified as false-negative cases. Cases of patients with none of the variables and no SAPI will be classified as true-negative cases. The remaining cases will be classified as false-positive cases. These values will be used to calculate point measures and 95 percent confidence intervals (CIs) for sensitivity, NPV and specificity of the decision instruments.

We will consider an instrument to be adequately validated if the lower confidence intervals for sensitivity and negative predictive value exceed 98%.

### Preliminary work

In this section we describe the development of our candidate criteria, our primary and secondary outcomes, and our sample size calculations.

### Assessing the inter-rater variability of criteria used in evaluating patients with blunt trauma for significant abdominal-pelvic injuries

**Purpose.** To determine the inter-rater variability of candidate criteria used to evaluate patients with blunt trauma for evidence of abdominal-pelvic injuries.

**Methods.** Physicians performed paired evaluations of patients with blunt trauma undergoing A/P CT imaging. Each physician independently determined whether each patient had any of the following characteristics: 1) abdominal pain or tenderness, 2) flank pain or tenderness, 3) pelvic pain or tenderness, 4) hip or iliac pain or tenderness, 5) midline lumbar spine or sacral pain, 6) abdominal distention, 7) abdominal or pelvic bruising, 8) abdominal or pelvic abrasion, 9) evidence of genital-urinary trauma, 10) abnormal alertness, 11) evidence of intoxication, 12) distracting painful injury, 13) hypotension, 14) tachycardia, 15) unstable vital signs, 16) low hemoglobin or hematocrit, 17) falling hemoglobin or hematocrit, 18) dangerous mechanism of injury, 19) abnormal plain radiography. Responses were compared using the Scott's pi statistic, with a point measure of 0.40 or greater interpreted to represent substantial inter-rater agreement.

**Results.** The physicians exhibited substantial inter-rater agreement for the following criteria: abdominal pain or tenderness ($\pi = 0.64$), flank pain or tenderness ($\pi = 0.43$), pelvic pain or tenderness ($\pi = 0.41$), hip or iliac pain or tenderness ($\pi = 0.49$), midline lumbar spine or sacral pain ($\pi = 0.47$), abnormal alertness ($\pi = 0.48$), evidence of intoxication ($\pi = 0.68$), distracting painful injury ($\pi = 0.43$), and hypotension ($\pi = 0.63$).

The physicians exhibited moderate or poor inter-rater agreement ($\pi < 0.40$) for abdominal distention, and abdominal or pelvic bruising or abrasion, tachycardia, unstable vital signs, and low or falling hemoglobin or hematocrit. The clinicians also exhibited poor inter-rater agreement on evidence of genital-urinary trauma ($\pi = 0.23$), but the exclusion of this criterion is mitigated by the fact that there were no patients who sustained significant genital-urinary trauma among our inter-rater assessment cohort.

**Conclusions.** We found substantial agreement between clinicians in their assessments of abdominal pain or tenderness, flank pain or tenderness, pelvic pain or tenderness, hip or iliac pain or tenderness, midline lumbar spine or sacral pain, abnormal alertness, evidence of intoxication, distracting painful injury, and hypotension. We were unable to obtain a meaningful assessment of genital urinary trauma due to its rarity among our cohort, and we found only moderate to poor agreement on the presence or absence of all other criteria.

### Defining significant injuries of the abdomen and pelvis

**Purpose.** To obtain consensus definitions of the significant injuries of the abdomen and pelvis that can serve in defining our primary and secondary outcome measures.

**Methods.** We employed a modified Delphi process involving a representative from emergency medicine, trauma surgery and trauma radiology from each of the three core centers. We provided each representative an initial list of individual abdominal and pelvic injuries. Participants were then asked to rate each injury as "clinically major," "clinically minor," or "clinically insignificant," and to provide any comments, questions or clarifications they deemed appropriate. Responses to the initial list of injuries were then tabulated, and the tabulated results, along with the respondent comments, were sent back to the respondents for further review.

We then provided reviewers with a final list of the individual abdominal and pelvic injuries, and asked them to provide a final classification based on their review of the tabulated results and reviewer comments. We based our final classification of the significance of each injury on the majority outcome ranking for each injury. Any injury patterns associated with tie votes were automatically assigned to the higher injury severity category.

**Results.** The respondents assigned "clinically major" importance to all injuries requiring intervention, including injuries to the bladder or ureter, bowel, diaphragm, genitals, hepato-biliary structures, hip, pancreas, pelvis, renal system, retroperitoneum, spine, spleen, and vasculature, including aorta, iliac or other vessels. Injuries to the spine that resulted in instability or neurological compromise were considered clinically major, even if they did not require intervention, as were injuries to the aorta. Respondents assigned "clinically minor" status to injuries that required observation but not intervention for the following areas: bladder or ureter, bowel, diaphragm, genitals, hepatobiliary structures, hip, pancreas, pelvis, renal system, retroperitoneum, spine (stable with no neurological compromise), spleen, and vascular injuries that do involve the aorta. Injuries that did not require intervention or observation were classified as "insignificant," with the exception of injuries to the aorta, and injuries to the spine that involve instability or neurological compromise.

**Conclusions.** Injuries of major clinical significance consist of all abdominal and pelvic injuries requiring intervention, as well as any injury to the aorta, and any injury to the spine associated with instability or neurological compromise. Injuries that require only observation, but no intervention, were considered clinically minor, provided they did not involve the aorta or spine, while injuries that required neither intervention nor observation were considered insignificant.

## Sample size calculations for derivation and verification studies

To be clinically reliable, the combined selective criteria must satisfy two requirements. First, patients identified as risk free by the decision instrument must never harbor injuries of major clinical significance. This is equivalent to requiring the instrument to have a 100% negative predictive value. Second, every patient with blunt trauma with an injury of major clinical significance must exhibit at least one of the risk criteria. This requirement implies that the sensitivity of decision instrument be 100%.

Verifying negative predictive value and sensitivity at an absolute level is not statistically possible with a finite sample size. However, it is possible to estimate limits for the true values of these proportions using exact statistical relationships. The lower confidence limit of a proportion (such as negative predictive value or sensitivity) is related to the total size of the study population. The quantitative relationship is expressed by the following equation from Fleiss [47]:

$$P_L = \frac{X}{X + (N - X + 1) \cdot F_{\frac{z}{2}, v_1, v_2}}$$

Where:

$P_L$ = Lower confidence limit for the proportion being studied (i.e. the lower confidence limit for the negative predictive value or sensitivity of selective criteria).

X = Number of times the event of interest is observed.

N = Total number of observations.

$\alpha$ = Statistical confidence value (Set to 0.05).

$v_1$ = Degrees of freedom for the numerator; $v_1 = 2 [N—X + 1]$

$v_2$ = Degrees of freedom for the denominator; $v_2 = 2 X$

$F_{\alpha/2, v1, v2}$ = One tailed value of the F-distribution with $v_1$ and $v_2$ degrees of freedom.

Validating a given $P_L$ to a 95% level of statistical certainty implies that $\alpha = 5\%$ (or $\alpha = 0.05$). The main hypothesis of this study states that selective criteria will detect all injuries of major clinical significance in patients with blunt trauma without missing any such injuries. Thus, for purposes of sample size calculations, for either negative predictive value or sensitivity, the event of interest (presence of one of the risk criteria) should occur with every observation. Hence, $X = N$.

The degrees of freedom for the numerator and denominator are then given by:

$\nu_1$ = Degrees of freedom for the numerator; $\nu_1 = 2 [N—X + 1] = 2 [N—N + 1] = 2$

$\nu_2$ = Degrees of freedom for the denominator; $\nu_2 = 2 X = 2 N$

The expression for the lower confidence limit then becomes:

$$P_L = \frac{N}{N + (N - N + 1) \cdot F_{0.025,2,2N}} = \frac{N}{N + F_{0.025,2,2N}}$$

Practical concerns regarding radiographic charges and radiation health effects suggest that selective criteria will still remain cost effective and reduce mortality and morbidity even when missing 2 out of every 100 injuries. This implies that negative predictive value need only be verified to a 98.0% lower confidence level, or equivalently, $P_L = 0.98$. Substituting this value in the lower confidence limit equation yields an algebraic equation for our injury population size, with a solution of $N = 317$. Thus, to confirm that the decision instrument has a negative predictive value of at least 98.0%, the study must enroll 317 blunt trauma patients who have none of the risk criteria, and none of whom have injuries of major clinical significance. Since approximately 10% of victims of blunt trauma are free of the risk criteria, this derivation would require a population of 3,170 patients with blunt trauma. Unfortunately, confirming that the instrument has 98.0% sensitivity requires studying 317 patients having injuries of major clinical significance. Major injuries occur in only 5% of cases of blunt trauma, which implies that this study will need to prospectively evaluate 6,340 patients with blunt trauma. Consequently, study size is driven by the requirements of sensitivity. Both the derivation and validation phases of the study will need enroll sufficient patient to satisfy the sensitivity requirement, thus the entire study will need to enroll a total of 12,680 patients with blunt trauma.

## Sample targets

Sensitivity = 95%; n = 125; N = 2,500; Total enrolled = 5,000

 Sensitivity = 96%; n = 157; N = 3,140; Total enrolled = 6,280

 Sensitivity = 97%; n = 211; N = 4,220; Total enrolled = 8,440

 Sensitivity = 98%; n = 317; N = 6,340; Total enrolled = 12,680

 Sensitivity = 99%; n = 637; N = 12,740; Total enrolled = 25,840

## Sample size calculations for the assessment of work-up bias

For our assessment of work-up bias we need to verify that occult injuries among patients who are not selected for A/P CT imaging is negligible, as evidenced by a prevalence below 1.0%. The equations of Fleiss again provide the quantitative relationships needed to determine the sample size for the work-up bias study, and confirm this low prevalence [47]. The equations indicate that a sample of 368 un-imaged patients will be needed to confirm that the prevalence of occult injuries does not exceed 1.0%. Thus our work-up bias assessment will need to enroll and follow 368 patients with blunt trauma who do not undergo CT imaging on their initial presentation.

### Planned secondary analyses

To be useful in changing practice, clinical decision tools should perform better than clinician gestalt in one of two ways: 1) increasing detection of injury (higher sensitivity than gestalt), or 2) decreasing test ordering without missing injuries (similar sensitivity and higher specificity than gestalt) [48]. As part of our pre-imaging clinical assessments we will ask physicians to estimate the likelihood that patients will have clinically significant injuries in A/P CT (gestalt). We will compare the screening performance of these gestalt estimates with the screening performance of our validated decision rules to ascertain whether the instruments improve on clinician gestalt.

As with our prior decision tools, we will conduct analyses to determine potential reductions in radiation exposure and secondary malignancies that can be expected under the use of our decision instruments [49]. We will also conduct cost efficacy impact analyses to determine potential resource savings with use of our A/P CT decision instruments [50].

Over the course of our study, we will collect detailed prospective information on a large cohort of patients with blunt trauma and with abdominal, pelvic and spine injuries. This detailed information provides us with a unique opportunity to examine multiple aspects of blunt abdominal trauma, including demographic characteristics, clinical presentations, injury patterns and management of patients with blunt trauma undergoing A/P CT imaging. We also plan to complete focused assessments of specific populations, including the young and the elderly, and analyses that address specific skeletal and organ injuries, including injuries to the spine, the pelvis, the liver, pancreas, spleen, and bowel.

## Concluding remarks

A decision instrument to guide A/P CT imaging of adult patients with blunt trauma could safely reduce CT imaging and associated radiographic charges, while also expediting trauma care and decreasing radiation exposure and its attendant risk of radiation-induced malignancy. Our detailed approach should enable us to successfully develop and validate an instrument that can readily be used in the clinical environment.

## Supporting information

**S1 File.**
(DOCX)

## Author Contributions

**Conceptualization:** Ali S. Raja, Robert M. Rodriguez, Malkeet Gupta, William R. Mower.

**Data curation:** Robert M. Rodriguez, William R. Mower.

**Formal analysis:** Ali S. Raja, Robert M. Rodriguez, William R. Mower.

**Investigation:** Ali S. Raja, Robert M. Rodriguez, Eric D. Isaacs, Lucy Z. Kornblith, Anand Prabhakar, Noelle Saillant, Paul J. Schmit, Sindy H. Wei, William R. Mower.

**Methodology:** Ali S. Raja, Robert M. Rodriguez, Malkeet Gupta, Eric D. Isaacs, Lucy Z. Kornblith, Anand Prabhakar, Noelle Saillant, Paul J. Schmit, Sindy H. Wei, William R. Mower.

**Project administration:** Ali S. Raja, Robert M. Rodriguez, Malkeet Gupta.

**Resources:** Ali S. Raja, Robert M. Rodriguez.

**Supervision:** Ali S. Raja, Robert M. Rodriguez, Malkeet Gupta, William R. Mower.

**Validation:** Ali S. Raja, Robert M. Rodriguez, William R. Mower.

**Visualization:** Ali S. Raja.

**Writing – original draft:** Ali S. Raja, Robert M. Rodriguez, William R. Mower.

**Writing – review & editing:** Ali S. Raja, Robert M. Rodriguez, Malkeet Gupta, Eric D. Isaacs, Lucy Z. Kornblith, Anand Prabhakar, Noelle Saillant, Paul J. Schmit, Sindy H. Wei, William R. Mower.

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
