## [Decision Letter · Decision Letter 0]

11 May 2022

PONE-D-22-06876Developing a Decision Instrument to Guide Abdominal-Pelvic Imaging of Blunt Trauma Patients: Methodology and Protocol of the NEXUS Abdominal-Pelvic Imaging StudyPLOS ONE

Dear Dr. Mower,

Thank you for submitting your manuscript to PLOS ONE. After careful consideration, we feel that it has merit but does not fully meet PLOS ONE’s publication criteria as it currently stands. Therefore, we invite you to submit a revised version of the manuscript that addresses the points raised during the review process.

It is an interesting and well done study. To my mind, the comments of reviewer 1 are important. Please, address his recommendations to improve your manuscipt.

We look forward to receiving your revised manuscript.

Kind regards,

Hans-Peter Simmen, M.D., Professor of Surgery

Academic Editor

PLOS ONE

Journal Requirements:

2. We noted in your submission details that a portion of your manuscript may have been presented or published elsewhere. (PLOS Medicine, full submission:PMEDICINE-D-22-00690)

Reviewers' comments:

Reviewer's Responses to Questions

**Comments to the Author**

1. Does the manuscript provide a valid rationale for the proposed study, with clearly identified and justified research questions?

Reviewer #1: Yes

Reviewer #2: Yes

2. Is the protocol technically sound and planned in a manner that will lead to a meaningful outcome and allow testing the stated hypotheses?

Reviewer #1: Yes

Reviewer #2: Yes

3. Is the methodology feasible and described in sufficient detail to allow the work to be replicable?

Reviewer #1: Yes

Reviewer #2: Yes

4. Have the authors described where all data underlying the findings will be made available when the study is complete?

Reviewer #1: Yes

Reviewer #2: Yes

5. Is the manuscript presented in an intelligible fashion and written in standard English?

Reviewer #1: Yes

Reviewer #2: Yes

6. Review Comments to the Author

You may also provide optional suggestions and comments to authors that they might find helpful in planning their study.

Reviewer #1: Thank you very much for the opportunity to review this interesting manuscript. The authors want to develop a decision instrument to guide abdomen-pelvis imaging in patients after a blunt trauma. They plan to do a multi-staged study, starting with a delphi consensus expert panel, second a truly observational study, and third a validating study.

I applause the authors to start such a large, multicenter project. However, I suggest to think about a few points:

- I don’t see a biostatistical expert in the author group, however a lot of biostatistical work has to be done (I also suggest our Editor to ask for statistical advice)

- Why do you want to combine pelvic and abdominal imaging? I understand this in potentially polytraumatized patients or in patients with an abdominal injury. In these cases we do a CT of the abdomen AND pelvis. However, we see e.g. a lot of elderly people with a blunt trauma to the pelvis / hip, where we just do a pelvic CT WITHOUT abdomen. I suggest to account for this.

- FAST: I suggest to add FAST to lines 86 / 89 / 326 / 542. There are still many emergency department starting with a FAST as an initial diagnostic work-up.

- The authors distinguish major / minor / insignificant injuries and want to draw conclusions based on these definitions. However, in my clinical daily practice we have some negative CTs. And I still would “always” do a CT in these cases (e.g. in selected trauma mechanisms such as patient is ejected out of a car, in selected injuries such as bilateral femoral fractures, in selected clinical findings such as an intubated patient). You mention this in supplement 1. Therefore, I think there is a group with a negative CT, however with a “good” reason to have been proceeded with a CT.

- What about adrenaline use (lines 325 and 540)

Reviewer #2: Statistically well refined concept, looking forward to the results in practice.

Maybe a detailed analysis of the role of FAST US vs MRI in the pediatric population could be of interest in the future?

7. PLOS authors have the option to publish the peer review history of their article (what does this mean?). If published, this will include your full peer review and any attached files.

Reviewer #1: No

Reviewer #2: No

---

## [Author Response · Author response to Decision Letter 0]

5 Jun 2022

Response to Reviewers

Query #1 - Journal Requirements 1 – A request to ensure that the manuscript meets PLOS ONE’s style requirements, including those for file naming.

Response #1 - We have revised the formatting and organization of the manuscript to meet PLOS ONE’s style requirements, including formatting for the main body of the manuscript, formatting of title, authors and affiliations, and formatting and numbering of references.

Query #2 - Concern that a portion of our manuscript may have been presented or published elsewhere.

Response #2 - No portion of this manuscript has been presented or published elsewhere. We did conduct pilot data collection at each of the participating centers to help us refine our protocols and ensure feasibility, and in the interest of efficiency, we plan to include this data in our final database. During this pilot work we asked clinicians to provide their estimates of the likelihood of injury for each patient, and the reason they elected to obtain imaging. This information is extrinsic to the derivation and validation of our decision instrument, but we do plan to prepare this material for separate peer-reviewed publication.

Query #3 – A request clarifying of our Data Availability statement.

Response #3 – We will provide repository information for our data at acceptance. Pending any last minute changes, we anticipate that our data will be available through the UCSF Datashare (Dash) repository at the following URL: https://datashare.ucsf.edu/stash/dataset/doi - pending.

Query #4 – A request to review our reference list for completeness and correctness.

Response #4 - We have reviewed and revised our reference list to ensure it is complete and correct and meets PLOS ONE’s formatting requirements. We have not changed any cited references, and our references do not included any retracted articles.

Query #5 – A reviewer request for biostatistical expert.

Response #5 – William R Mower, MD, PhD, the director of the NEXUS group and communicating author for the manuscript, holds a PhD in biomathematics and has over 30 years of experience in statistical research design. He developed the analytic plan and sample estimates for the project and will serve as the study statistician.

Query #6 - Concern that we are combining pelvic and abdominal imaging into a single decision instrument.

Response #6 - Our project will develop an instrument for combined abdominal-pelvic imaging, as this is the form of imaging that is frequently used in the evaluation of multi-trauma victims. However, the final rule will also apply to isolated abdominal imaging, or isolated pelvic imaging. This is a reflection of the fact that a patient who is designated as low risk by our instrument will have negligible risk of either abdominal or pelvic injury, and can safely be excluded from abdominal-pelvic imaging, isolated abdominal imaging, or isolated pelvic imaging. 

We have included this clarification in the “General Design” section of the manuscript (lines 147 to 152 in the revised manuscript).

Query #7 – A Request to include focused assessment with sonography in trauma (FAST) in our project.

Response #7 – We recognize the importance of FAST exams in the assessment of blunt trauma patients, and have included FAST scan results as one of our candidate criteria (Supplement 1 presents our data collection instrument and contains a full listing of candidate variables). We have modified our “Introduction” to acknowledge that FAST scanning is among the imaging modalities commonly used in abdominal imaging of blunt trauma patients (lines 94 to 97 of the revised manuscript – see also Supplement 1 – Abdominal/Pelvic CT Clinical Questionnaire).

Query #8 – A reviewer is concerned that there may be cases where a clinician may want to obtain imaging regardless of the classification by the decision rule.

Response #8 - We agree that there may be instances where clinicians may want obtain imaging regardless of classification by a decision rule. Our recommendation would be to not apply the decision tool if you plan to obtain imaging regardless of classification. The goal of the tool is to help clinicians identify cases that have such low risk of injury that they may be safely excluded from imaging. Our goal is not to force imaging decisions on clinicians.

We have not changed the manuscript in response to this concern. However, it is worth noting that each of the cases described by the reviewer would be identified by our current criteria (ejected out of a car is a rapid deceleration mechanism, bilateral femoral fractures would represent distracting injuries, and intubation would exclude low risk classification for several reasons, including physiologic instability, abnormal level of alertness, and inability to complete and assessment of all criteria, which automatically excludes patients from low risk classification).

It is impossible to develop decision tools that anticipate every possible scenario, and it is possible that our final rule may occasionally misclassify patients with important injuries. If the frequency of missed injury cases cause the tool’s sensitivity to fall below our pre-specified rate of 98%, then the harm from the rule (morbidity and mortality from missed injuries) will exceed the benefits (decreased imaging with a decrease in radiation induced malignant transformation) and we would fail in our efforts to develop a decision instrument. This failure would still be informative and would provide important information for future research.

Query #9 – A request to include adrenaline use as a criterion.

Response #9 - While the use of adrenaline is likely associated with significant injuries, it is an intervention, and not a sign or symptom of injury. Adrenalin is most likely to be used in cases that exhibit hypotension, and hypotension is already included among our potential criteria.

We have mad no changes to the manuscript related to this concern. 

Query 10 – Requesting a detailed analysis of the role of FAST US vs MRI in the pediatric population in the future.

Response #10 – We agree that FAST US and MRI play particularly important roles in the pediatric population, and that future assessments of these modalities would be important in defining care. However, the assessment of these modalities is beyond the scope of this project.

We have mad no change to the manuscript related to this concern.

---

## [Editor Report · Decision Letter 1]

23 Jun 2022

Developing a Decision Instrument to Guide Abdominal-Pelvic Imaging of Blunt Trauma Patients: Methodology and Protocol of the NEXUS Abdominal-Pelvic Imaging Study

PONE-D-22-06876R1

Dear Dr. Mower,

We’re pleased to inform you that your manuscript has been judged scientifically suitable for publication and will be formally accepted for publication once it meets all outstanding technical requirements.

Kind regards,

Hans-Peter Simmen, M.D., Professor of Surgery

Academic Editor

PLOS ONE
---

## [Editor Report · Acceptance letter]

14 Jul 2022

PONE-D-22-06876R1 

Developing a Decision Instrument to Guide Abdominal-Pelvic Imaging of Blunt Trauma Patients: Methodology and Protocol of the NEXUS Abdominal-Pelvic Imaging Study 

Dear Dr. Mower:

I'm pleased to inform you that your manuscript has been deemed suitable for publication in PLOS ONE. Congratulations! Your manuscript is now with our production department. 

Kind regards, 

on behalf of

Dr. Hans-Peter Simmen 

Academic Editor

PLOS ONE